# Short-term hormonal modulation with mifepristone does not induce oncogenic changes in the endometrium of *BRCA1/2* pathogenic variant carriers

Martin Widschwendter [1,2,3,4,5,10], Chiara Herzog [2,3,10], Mohammed Fatih Rasul[1,6],
Nageswara Rao Boggavarapu[1], Elisa Redl [2,3], Deborah Utjés[1,7], Angelique Flöter Rådestad[1,8],
Kristina Gemzell- Danielsson [1,7,10] & Twana Alkasalias [1,9,10] ✉

## Abstract

**Background** Progesterone receptor antagonists such as mifepristone have emerged as candidates for breast cancer prevention, particularly in high-risk populations such as *BRCA1/2* pathogenic variant carriers. However, their impact on endometrial safety remains insufficiently characterized, raising concerns about unopposed oestrogen stimulation in the setting of impaired DNA repair. This study reports secondary outcomes evaluating short-term endometrial effects of mifepristone in this high-risk population.

**Methods** We previously conducted a randomized, double-blind, placebo-controlled trial (NCT01898312) involving 45 premenopausal women with *BRCA1/2* pathogenic variants. Participants received mifepristone (50 mg every other day, n = 30) or a non-hormonal comparator (n = 15) for three months. Here we present secondary outcomes from the trial: Paired endometrial biopsies from a subset of 14 participants were analysed using transcriptomic, DNA methylation, and cell-type deconvolution methods. Statistical comparisons were performed using paired and unpaired Wilcoxon tests.

**Results** Here we show that mifepristone induces amenorrhea in all treated participants without increasing epithelial cell proportions, the compartment most relevant to endometrial carcinogenesis. Multi-omics analyses reveal no molecular signatures consistent with oncogenic transformation. DNA methylation and gene expression indices associated with endometrial cancer remain stable after treatment, even after adjusting for age and cell composition.

**Conclusions** Short-term mifepristone exposure does not produce molecular changes linked to endometrial carcinogenesis in *BRCA1/2* pathogenic variant carriers. These findings provide important safety data for the future development of progesterone receptor modulators in cancer prevention. Long-term studies are needed to confirm these observations.

## Plain language summary

Women who have inherited different versions (mutations) of the *BRCA1* or *BRCA2* gene are more likely to get breast cancer. Medicines that block the activity of the hormone progesterone, such as mifepristone, are being evaluated to determine whether they can be taken to prevent development of breast cancer. However, it is important to know what impact taking these medicines has on the lining of the womb (endometrium), because changes in the endometrium can result in development of cancer. We gave mifepristone or a placebo to women with mutations in *BRCA1* or *BRCA2* for three months and examined tissue samples from the womb. We found that mifepristone caused periods to stop but did not increase cell growth or result in changes usually seen when cancer develops. These results suggest that short-term use of mifepristone is safe for the womb, but longer studies are needed to evaluate longer term effects.

The biology and function of the endometrium are regulated by steroid hormones, and any imbalance in hormone signaling is associated with an increased risk of endometrial diseases. These conditions include endometrial hyperplasia, heavy menstrual bleeding, subfertility, endometriosis, and endometrial cancer (EC)[1–3]. Estrogen (E) stimulates endometrial cell proliferation, whereas progesterone (P) inhibits their growth and promotes cell differentiation[4]. Consequently, elevated levels of E relative to P are associated with an increased risk of developing EC[4,5]. Globally, EC remained

---

among the most common gynecologic malignancies globally, with incidence and mortality continuing to rise. Recent estimates report over 420,000 new cases worldwide and nearly 98,000 deaths annually, reflecting a persistent upward trend[6]. Although some genetic factors have been linked to EC, hormonal imbalance remains the primary risk factor for its development[4,5].

Progesterone regulates cell growth in hormone-responsive tissues via various mechanisms. For instance, in breast tissue, P promotes cell proliferation via the RANK pathway and acts as a pro-tumorigenic agent[7,8] Women carrying pathogenic *BRCA1* variants exhibit higher P levels, which increases the risk of triple-negative breast cancer (TNBC) by enhancing the proliferation of PR-negative luminal progenitor cells through the RANK pathway[9–17]. In contrast, in the endometrium, P acts as a tumor suppressor by countering the pro-tumorigenic effects of E[12,14,18,19]. This protective role of P is critical in preventing the unopposed effects of E, where prolonged E stimulation without adequate P can lead to endometrial hyperplasia and an increased risk of EC[20–22]. Considering the protective role of P in the endometrium, mifepristone (RU 486), a progesterone receptor antagonist or modulator (PRM), could theoretically result in unopposed E exposure, raising concerns about the risk of endometrial carcinogenesis[23–26].

Mifepristone has diverse applications, including its use in abortion, management of early and late pregnancy loss, and cervical priming prior to surgical abortion[27–29]. It has also been employed in the treatment of endometriosis, adenomyosis, Cushing's syndrome, leiomyoma, and for inducing amenorrhea[30–34]. Furthermore, mifepristone has been investigated as a safe, long-term, oestrogen-free contraceptive pill without an associated increase in hyperplasia risk[35,36]. However, mifepristone has been linked to distinct histological changes in the endometrium, known as progesterone receptor modulator-associated endometrial changes (PAEC)[37]. These changes include cystic dilatation of the endometrial glands, irregular glandular architecture, and non-decidualized stroma, which are not observed together under physiological conditions. The potential implications of PAEC, particularly in women with *BRCA* pathogenic variant (PV) mutations, are not yet fully understood.

Given the key role of P in driving TNBC in women with PV germline *BRCA* mutations, we designed a clinical trial to investigate the molecular changes in breast tissue induced by mifepristone treatment and to explore its potential as a cancer preventive measure in this population. However, an important concern is the potential impact of mifepristone on the endometrium. The combination of hormonal modulation (progesterone signaling alteration) and genetic vulnerabilities in DNA repair mechanisms can increase the risk of endometrial abnormalities or malignancies. Understanding these risks is critical to ensure that cancer prevention strategies for *BRCA* PV carriers are both effective and safe.

Here we show that short-term mifepristone treatment induces amenorrhea without increasing epithelial cell proportions or causing molecular changes linked to endometrial carcinogenesis. Multi omics analyses reveal stable DNA methylation and gene expression profiles, supporting the endometrial safety of short-term mifepristone exposure.

## Methods
### Study overview and tissue sampling
This study reports the secondary outcomes of a randomized double-blind placebo-controlled trial registered under EudraCT Number: 2012-003703-35, NCT01898312. Women with PV of *BRCA1/2* mutations were recruited during scheduled surveillance appointments at the gynecological outpatient unit of the Division of Hereditary Cancer, Karolinska University Hospital (Fig. 1) All participants were unaffected *BRCA1/2* PV carriers with no previous history of risk reducing mastectomy or diagnosis of breast or other cancers. The included women were randomized to a 3-month treatment with mifepristone (50 mg; Mifegyne, Exelgyn, Paris, France) or a non-hormonal visually indistinguishable comparator (TrioBe, which is a cocktail of three different types of vitamin B) administered orally every second day. The 50 mg every other day regimen was selected based on previous clinical studies demonstrating its efficacy in inducing amenorrhea without

significant adverse effects[34,38,39]. This dosing schedule also minimizes cumulative exposure while maintaining pharmacological activity. Endometrial biopsies were obtained before the start of treatment and again after three months of treatment. All baseline biopsies for both mifepristone and vitamin B groups were collected during the luteal phase, as determined by cycle day. For post-treatment biopsies, women treated with mifepristone were amenorrheic, whereas women in the vitamin B group continued menstruating; therefore, their post-treatment samples were also obtained during the luteal phase. To further minimize variability, luteal phase samples were stratified into early, mid, and late luteal sub-phases, although no significant differences were observed across these sub-phases. Women were instructed to use barrier methods for contraception during the treatment period.

Owing to the relatively low rate of *BRCA* mutations and the fact that women could serve as their own controls, randomization was implemented in a 2:1 ratio. To compensate for loss to follow-up and inadequate biopsy material, 30 women were included in the *BRCA* group, and 15 were randomized to receive vitamin B treatment. During screening, the women underwent a general examination, including blood pressure, height and weight, hematology, kidney and liver function tests, and thyroid function. A gynecological examination including chlamydia test, pap smear and vaginal ultrasonography with measurements of endometrial thickness was conducted.

Tablets were purchased, labeled, and distributed via the University Hospital Pharmacy. Tablets were handed out in a special box, for one week at a time, and women were instructed to return to the WHO-clinical trial center at Karolinska University Hospital to receive new tablets every week during the treatment period. Compliance was checked weekly when the women returned to obtain new tablets. Empty boxes or blisters and any remaining tablets were returned to the site by the women at each visit. The women reported any side effects recorded during the three months of treatment. During the last week of treatment, a general and gynecological examination was performed as before the study. As the trial was part of a larger trial in which we also studied the effect of mifepristone on breast tissue, endometrial biopsies were collected at the gynecological outpatient clinic. Fresh biopsies were transferred to the laboratory and stored at −80 °C until processing.

The inclusion criteria for the study required participants to be premenopausal women aged ≥18 years who carry at least one PV of *BRCA1/2* mutations. They consented to undergo breast biopsy under local anesthesia before and after the three-month treatment period. Endometrial biopsies were performed on the same day as the breast biopsies at the gynecological outpatient clinic. Participants also needed to be in good general health, have regular menstrual cycles lasting between 25 and 35 days, and be willing to participate after providing their informed consent. The endometrial changes were assessed using descriptive statistics.

### DNA and RNA extraction from endometrial tissue biopsies
RNA and DNA from 14 paired samples were extracted using the Zymo Quick-DNA/RNA Microprep Plus Kit with minor modifications, leveraging beads alongside proteinase K and the digestion buffer to achieve better digestion and homogenization efficiency. The Kit and the manufacturer's protocols were followed. RNA and DNA quantification was conducted using the Qubit RNA High Sensitivity Assay Kit and 1X dsDNA HS Assay Kit, respectively (Thermo Fisher Scientific).

### Methylation data generation and quality control
Extracted DNA (250 ng) was bisulfite converted and subsequently 200 ng were profiled using the Illumina Human Methylation EPIC array, version 1. Raw IDAT files were processed using the eutopsQC (github.com/chiaraherzog/eutopsQC) pipeline, as previously described[40].

### RNA sequencing, processing, and quality control
cDNA library preparation and sequencing. cDNA libraries for next-generation sequencing were constructed using RNA extracted from 14

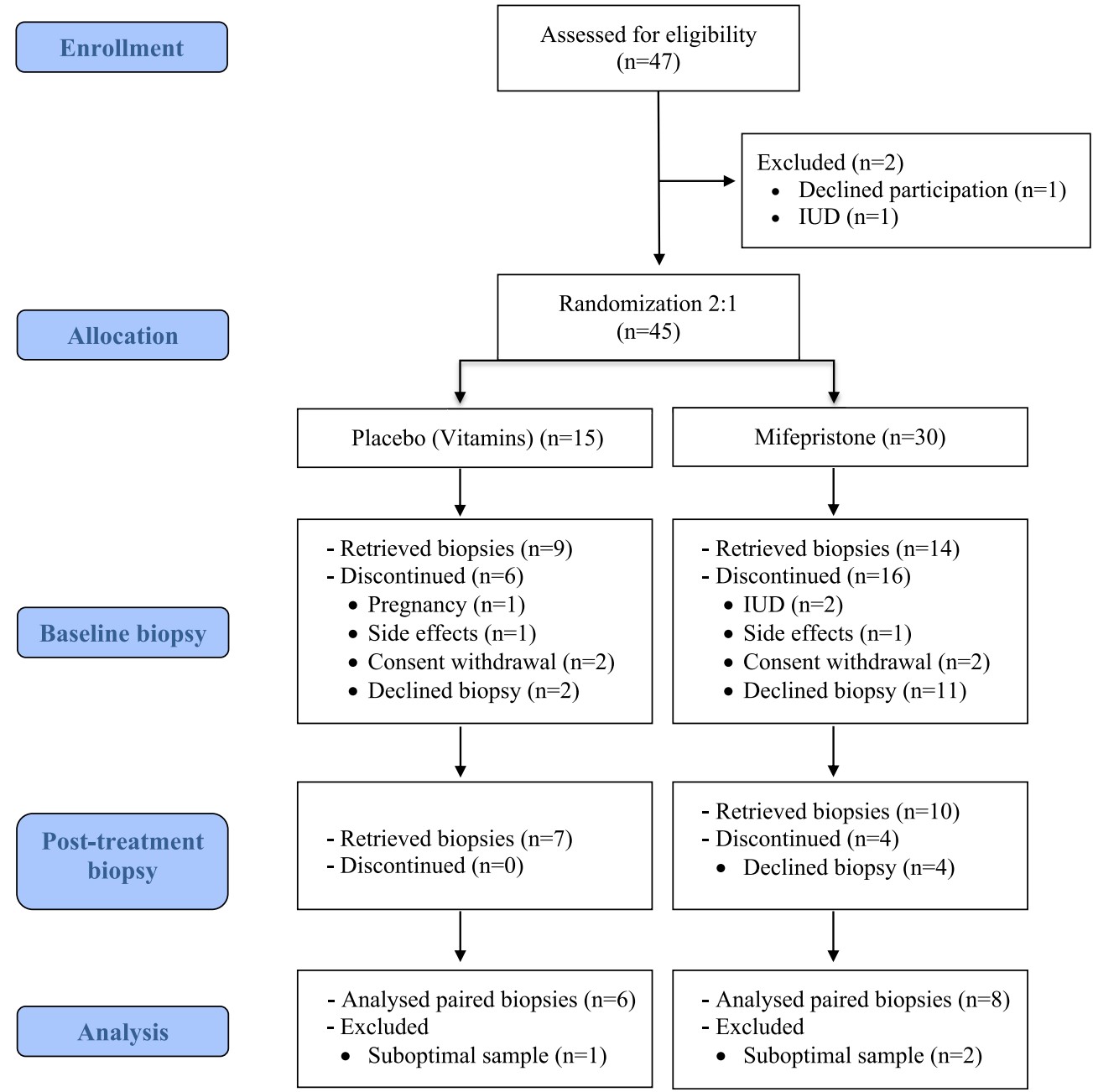

**Fig. 1 | Study flow chart illustrating the process of patient enrollment, allocation, randomization, biopsies, and analysis.** Sample sizes: $n = 45$ participants randomized ($n = 30$ mifepristone, $n = 15$ control); $n = 14$ paired endometrial biopsies analyzed for molecular endpoints.

paired samples obtained before and after the mifepristone treatment. The Smart-seq2 protocol was used as previously described[41]. For each sample, 1 ng of RNA, quantified using a Qubit, was used as the starting material for library construction.

cDNA tagmentation was performed using a Nextera XT Kit (Illumina), followed by adapter and index primer addition according to the manufacturer's recommendations. Illumina Nextera XT Index Kit (catalog no. FC-131-1002) was used to multiplex samples for sequencing. The sequences of the oligonucleotides used were as follows: Oligo-dT30VN (5′-AAG-CAGTGGTATCAACGCAGAGTACTTTTTTTTTTTTTTTTTTTTTTT TTTTTTTVN-3′), Template Switching Oligo (5′-AAGCAGTGGTAT-CAACGCAGAGTACATrGrG+G-3′), and IS PCR Oligo (5′-AAG-CAGTGGTATCAACGCAGAGT-3′).

Post-Nextera reaction products were purified using AMPure XP beads (Beckman Coulter, USA) at a 1:1 bead-to-sample ratio. The purified libraries were quantified using a Qubit Flex Fluorometer with a 1X dsDNA HS Assay Kit (Life Technologies). Quality control was conducted using a High-Sensitivity DNA chip on a 2100 Bioanalyzer (Agilent Technologies). Subsequently, 10 ng of DNA from each library was pooled and sequenced on an Illumina NovaSeq 6000 instrument using a paired read 2× 150 cycle set up at Novogene Limited (U.K).

**Data processing and analysis.** RNA-seq data analysis was performed using the Partek Flow Genomic Analysis Software (Partek Inc., St. Louis, MO, USA). FASTQ files were processed to remove contaminants, including ribosomal and mitochondrial DNA, using the Bowtie 2 aligner. The standard Nextera Transposase adapter (CTGTCTCTTATACA-CATCT) was trimmed from raw reads. Filtered reads were aligned to the Hg38 genome using STAR aligner with default settings. The total alignment rates ranged from 95 to 99%, with unique alignment rates of 80–92%

and an average Phred quality score of 34 per base post-alignment. Filtered alignments were quantified against hg38 Ensembl transcript release 100. Gene features were filtered to retain those with at least one count in 80% or more of the samples, resulting in 26,581 genes (78%) that met this criterion. Differential expression analysis was performed using the DESeq2 tool in the Partek platform (Supplementary Fig. 1).

## Development of a methylation-based EC index

We leveraged DNA methylation data from The Cancer Genome Atlas (TCGA) UCEC project for the development of a methylation-based index to discern normal endometrial tissue from EC, as previously demonstrated for breast cancer by Barrett et al.[40]. Harmonized TCGA-UCEC methylation data were obtained using TCGA Biolinks (version 2.30.0). The resulting dataset, consisting of 484 samples, including 46 normal adjacent tissues and 438 EC tissues, was randomly split into training (70%) and testing (30%) sets. After adjusting for age and estimated immune cell proportion, we identified 180,874 significantly differentially methylated CpGs, 49,179 of which remained significant after adjusting for the false discovery rate. Moreover, using linear models for cases and controls only, we estimated the methylation difference (delta beta) in pure epithelial cell populations, that is,

at ic (immune cell composition) = 0, based on the intercepts of each model. CpGs were ranked by the largest estimated difference (delta-beta) in epithelial cells, and the top 30,000 CpGs with the largest estimated difference in methylation between cancer and control tissues in epithelial cells were leveraged for model training using the R package glmnet (version 4.1.8). We explored the use of ridge, elastic net, or lasso-regularized penalization (alpha = 0, 1, or 0.5, respectively) with increasing numbers of CpGs as input features (100, 500, 1000, and up to 30,000 in steps of 1000). The models were then compared using the area under the curve and calibration (slope and intercept), which were derived from the val.prob function of the rms package (version 6.8.1). Ridge penalization with 8000 top-ranked CpGs yielded the highest area under the curve and the highest calibration (slope and intercept) and was subsequently used as the final index (Supplementary Data 2). The index was validated in a hold-out TCGA testing set, where it achieved a perfect distinction between EC and normal tissue (AUC = 1), and was subsequently applied to the dataset generated in this study.

## Development of an RNA-seq based EC index

We used RNA-seq count data from The Cancer Genome Atlas (TCGA) UCEC project for differential gene expression analysis of EC versus solid

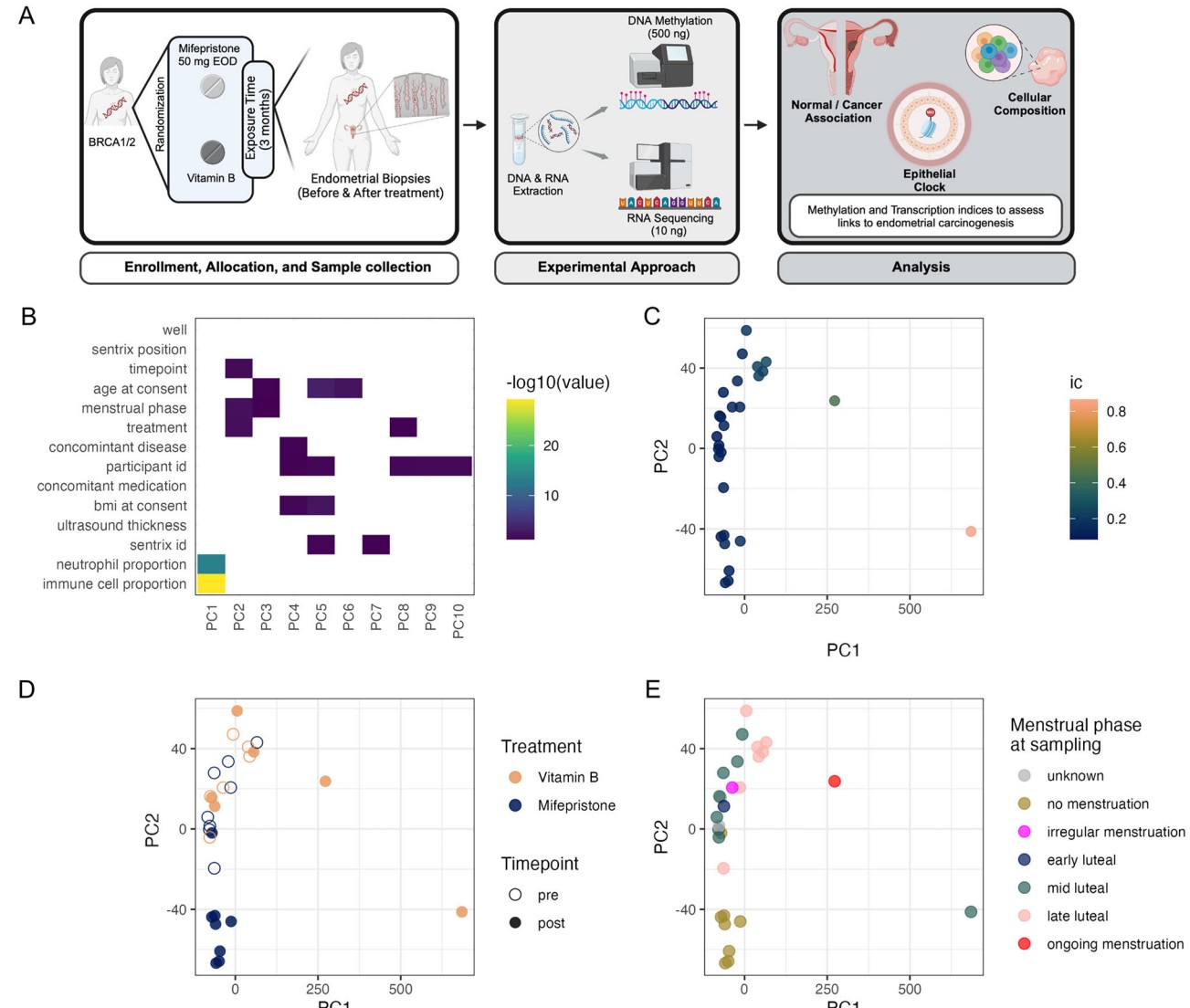

**Fig. 2 | Study overview and DNA methylation data quality control. A** Study overview(Created with BioRender.com). **B** Heatmap of principal components of the top 30,000 variable CpG sites in the DNA methylation dataset with technical and biological factors. **C** Principal components colored by immune cell proportion (ic), **D** treatment and timepoint, and **E** by menstrual phase at sampling. Sample sizes: $n = 14$ paired endometrial samples (pre- and post-treatment) analyzed for methylation profiling.

normal tissues and for index development. Raw data for primary solid tumors and normal adjacent tissues were obtained using TCGAbiolinks (version 2.30.0), which included 35 normal tissues and 541 tumor tissues. EntrezIDs were converted into gene symbols using the R package org.H-s.eg.db (3.18.0), merging the counts from several transcripts of the same genes. The resulting dataset consisted of 576 samples with expression values of 36,152 unique gene symbols, each split randomly into a training (80%) and testing (20%) sets, ensuring that samples from the same individual were within the same set (training or testing, respectively). Differential gene expression analysis was conducted on the training set using DESeq2 (1.42.0) on genes with a read count of at least 10 in more than 80% of the samples, retaining 14,318 genes, of which 8465 were significantly differentially expressed in the EC tissue compared with the control tissue ($P < 0.01$). While the same algorithm for training was used to derive methylation and expression indices (i.e., glmnet), for the expression data, we did not perform hyperparameter tuning and instead used the log2- and median of ratio-normalized counts ($n + 1$) of all significantly differentially expressed genes as inputs for elastic net regression. Sixty-four genes were retained in the final classifier index (Supplementary Data 3) and were validated by hold-out testing. Finally, the index was applied to the log2 and median of the ratio-normalized counts of genes ($n + 1$) in the dataset generated as part of this study.

### Statistical analysis and assessment of index changes
Statistical analysis was performed in the R computing environment, version 4.3.1 (2023-06-16). No formal sample size calculation was performed for the secondary outcomes. Summary statistics were generated using the gtsummary version 1.7.2. Graphs were generated using ggplot2 version 3.5.0, and ComplexHeatmap, version 2.18.0. Paired Wilcoxon tests were used to compare values at baseline and at the end of treatment, whereas unpaired Wilcoxon tests were used to compare changes from baseline between the respective treatment groups. Statistical tests were performed using the ggpubr package, version 0.6.0, for ggplot2. The statistical analysis included two comparisons: (1) between the treated group and the control (vitamin B) group to assess the overall treatment effect and (2) within individuals by comparing baseline and end-of-treatment measurements, acknowledging that this approach allows evaluation of changes over time within the same participant. The codes for the analyses and figures are provided on GitHub (https://github.com/chiaraherzog/MifepristoneEndoMethyl), and have been archived with (https://doi.org/10.5281/zenodo.18219939)[42].

### Ethical approval and data access
The trial was conducted in compliance with the International Conference on Harmonization Guidelines for Good Clinical Practice, principles of the Declaration of Helsinki, and national regulations and directives. The protocol and amendments (W2012B, EudraCT Number: 2012-003703-35) were approved by the institutional review board of the Karolinska Institutet and applicable regional regulatory agencies. Written informed consent was obtained from all participants prior to their enrollment in the study.

Publicly available DNA methylation and gene expression data from The Cancer Genome Atlas (TCGA) UCEC project were accessed via TCGAbiolinks (version 2.30.0). As the data are de-identified and publicly available, no additional consent was required for secondary analysis.

## Results
### Mifepristone treatment resulted in amenorrhea and induced a unique endometrial methylation signature
In this randomized clinical trial, women with PV *BRCA* mutations were treated with either mifepristone or a control (vitamin B) for three months (Figs. 1 and 2A). Baseline characteristics, such as age at consent, BMI, and ultrasound-measured endometrial thickness, are summarized in Table 1, and Supplementary Data 1. The first patient was recruited from 2015-02-12 and the last patient was recruited from 2019-10-10. The last follow-up visit

**Table 1 | Participant characteristics**

| Characteristic | Vitamin B | | Mifepristone | |
|---|---|---|---|---|
| | Pre $N$ = 6[a] | Post $N$ = 6[a] | Pre $N$ = 8[a] | Post $N$ = 8[a] |
| Age at consent | 37.0 (33.0, 43.0) | 37.0 (33.0, 43.0) | 38.5 (35.5, 42.5) | 38.5 (35.5, 42.5) |
| BMI | 21 (20, 42) | 21 (20, 42) | 23.3 (21.9, 26.4) | 23.3 (21.4, 26.8) |
| unknown | | | 1 | 1 |
| Menstrual phase | | | | |
| early luteal | – | 1 (17%) | – | – |
| mid luteal | 3 (50%) | 1 (17%) | 4 (50%) | – |
| late luteal | 2 (33%) | 2 (33%) | 3 (38%) | – |
| ongoing menstruation | – | 1 (17%) | – | – |
| irregular menstruation | 1 (17%) | – | – | – |
| no menstruation | – | – | – | 8 (100%) |
| unknown | – | 1 (17%) | 1 (13%) | – |
| Ultrasound endometrial thickness (mm) | 9.00 (5.90, 9.00) | 6.00 (4.00, 13.00) | 8.5 (8.0, 12.0) | 10.5 (6.0, 13.0) |

[a]Median (Q1, Q3); $n$ (%).

was 2020-01-13. All the participants (100%) in the mifepristone group experienced amenorrhea during the follow-up. Ultrasound measurements demonstrated no significant difference in endometrial thickness changes between the mifepristone and control (vitamin B) groups pre-and post-treatment. The median thickness in pre- and post- mifepristone treatment cohorts was of 8.5 mm (IQR 8.0, 11.5) 10.5 mm (IQR: 6.5–13.0), respectively, whereas the pre- and post- treatment thickness in the vitamin B group was of 9.00 mm (IQR: 6.68, 9.00) and 6.00 mm (IQR: 4.25, 11.50) respectively.

Methylation data quality control revealed distinct clustering of samples based on the treatment group (Fig. 2B–E). As expected, the most pronounced effect on the principal components was exhibited by sample composition. However, principal components 1–3 also exhibited significant associations with the menstrual phase, treatment, and ultrasound thickness. For downstream analysis, we ruled out any sample mismatches via single nucleotide polymorphism (SNP) probe analysis, excluding one individual with an SNP mismatch between the pre- and post-samples (Supplementary Fig. 3).

### Mifepristone drives the cellular composition of endometrial tissue
Epithelial cells proliferate in response to hormonal signaling and are considered EC precursors[43]. Given this critical role, we specifically assessed the impact of mifepristone on the epithelial cell population within endometrial tissue. To investigate this effect, we used DNA methylation data for accurate cellular compositional inference and applied (hierarchical) EpiDISH, an algorithm designed to infer the relative proportions of epithelial cells, fibroblasts, and various subtypes of immune cells in each sample, both before and after treatment. When comparing the pre- and post-treatment samples within the mifepristone group, our analysis (Fig. 3) showed a small decrease in the epithelial cell proportion from 55% to 51%, although this decrease was not statistically significant ($P = 0.078$), fibroblast proportion increased significantly from 31% to 36% ($P = 0.0078$). When comparing the differences in delta (Δ post-pre values) between the mifepristone and control (vitamin B) groups, no significant difference was observed in the epithelial cell proportions ($P = 0.75$), whereas the proportion of fibroblasts was significantly higher in the mifepristone group than in the control group ($P = 0.029$). No notable alterations were observed in the other cell types in either the mifepristone or the control (Vitamin B) groups. Furthermore,

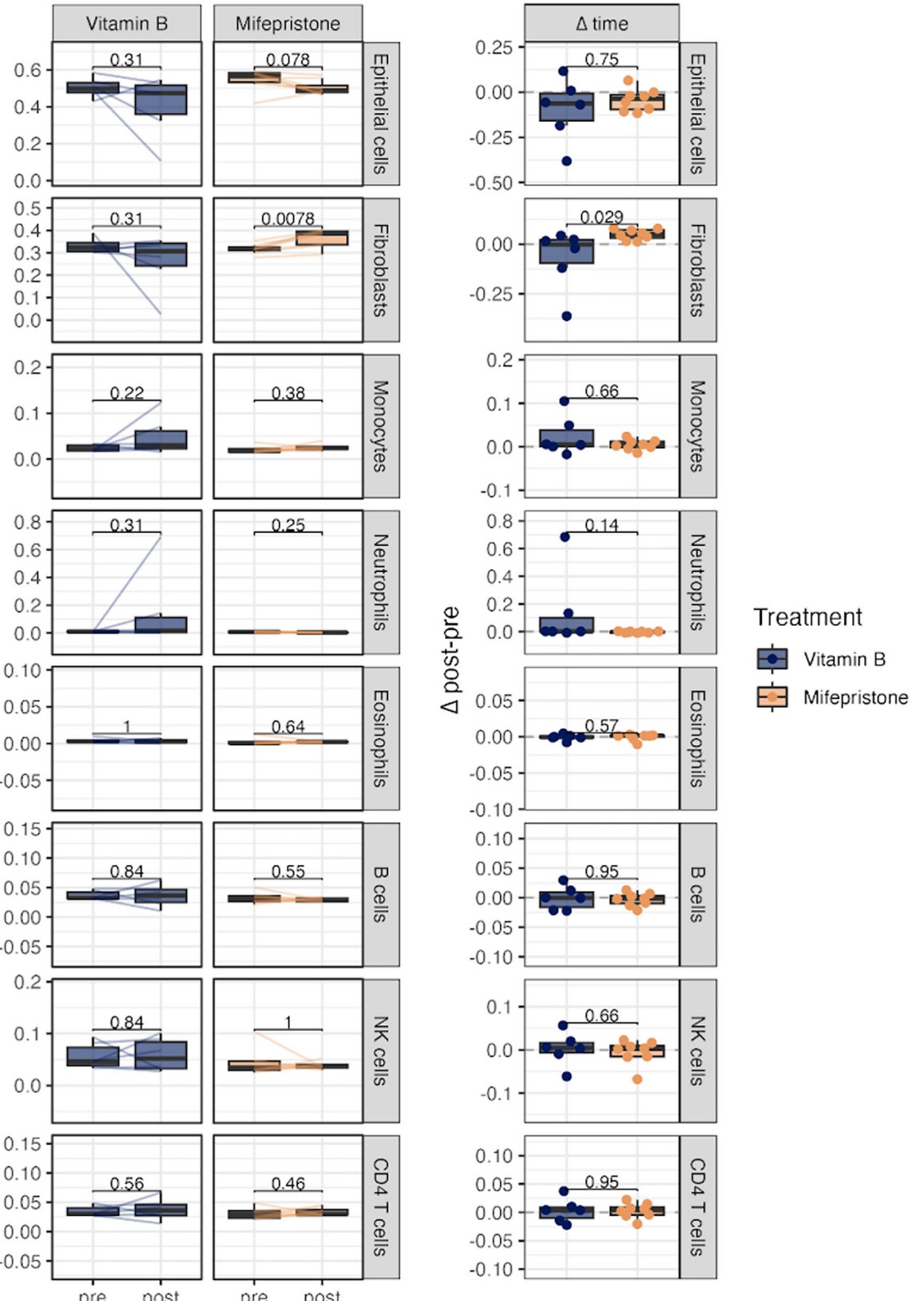

**Fig. 3 | Mifepristone-induced changes in DNA methylation-based inferred cell type proportions.** Cell type composition was estimated using the hierarchical EpiDISH algorithm, showing paired values before and after Vitamin B or Mifepristone treatment. Δ time indicates within-individual change from baseline to follow-up biopsy. Plots for CD8T cells are excluded as they were not detected in any sample. *p* values for longitudinal comparisons are calculated using two-sided paired Wilcoxon tests, while Δ time comparisons are compared using two-sided unpaired Wilcoxon tests. Sample sizes: *n* = 14 paired samples (mifepristone group: *n* = 8; control group: *n* = 6).

## DNA methylation data

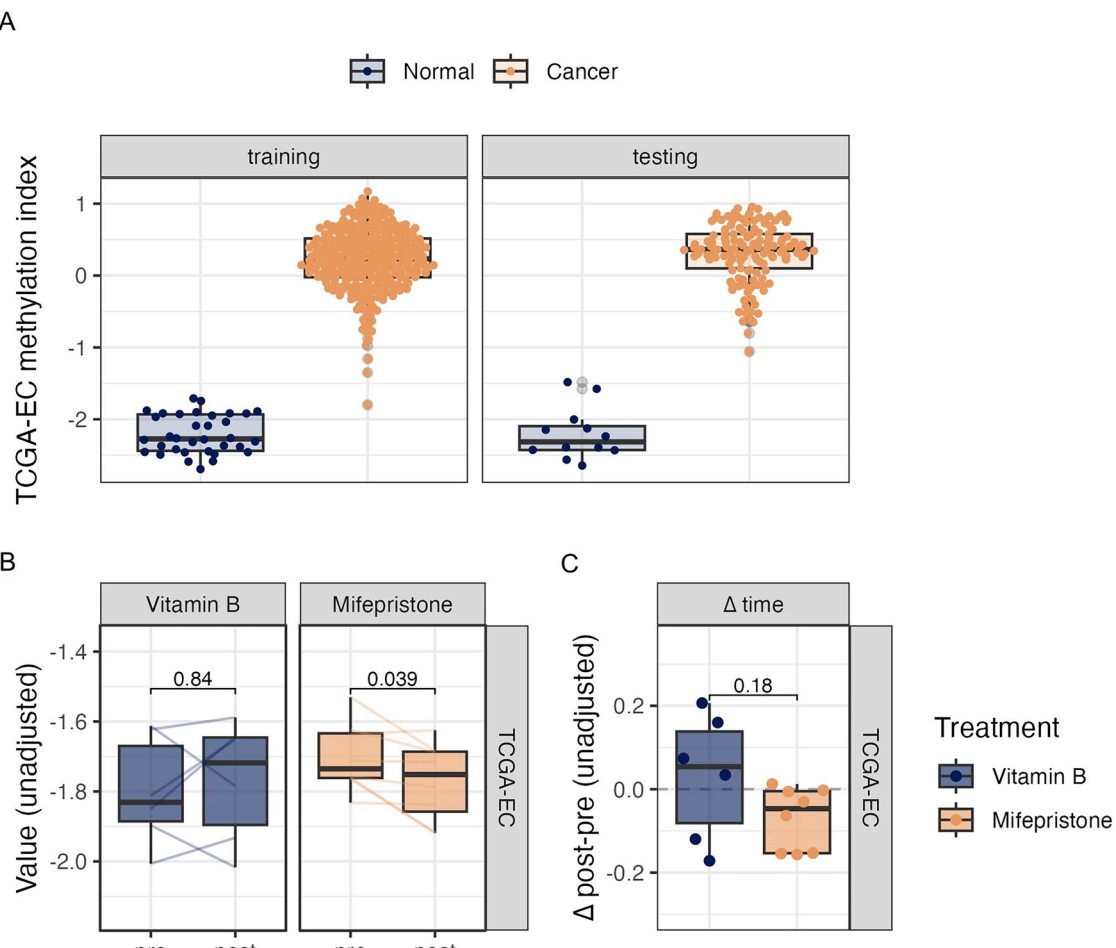

**Fig. 4 | Changes in cancer-indicative DNA methylation biomarkers upon mifepristone treatment. A** DNA methylation-based Endometrial Cancer Tissue index. Values of the methylation-based composite TCGA-EC index, consisting of 8000 CpGs, in TCGA data from the UCEC project by training and testing set. **B** Paired pre and post values of cancer-indicative biomarkers associated with biomarkers of endometrial cancer based on DNA methylation, trained using TCGA data (TCGA-EC methylation index). $p$ values are derived from two-sided paired Wilcoxon tests. **C** Change in biomarker values from baseline (Δ time) in Vitamin B and Mifepristone-treated individuals. $p$ values derived from two-sided unpaired Wilcoxon tests comparing Mifepristone to Vitamin B. Sample sizes: $n = 14$ paired samples (mifepristone group: $n = 8$; control group: $n = 6$).

CD8T cells were undetectable in all samples and were therefore excluded from the analysis.

### Mifepristone alters methylation index related to endometrial carcinogenesis

DNA methylation signatures are powerful tools that can detect the presence of EC and predict its risk, as methylation changes are closely associated with carcinogenic progression in endometrial tissues[44]. To investigate the molecular effects of mifepristone on endometrial tissue and its potential correlation with carcinogenesis, we utilized methylation data from TCGA Uterine Corpus Endometrial Carcinoma Project (UCEC) to derive a methylation-based classifier consisting of 8000 sites, termed TCGA-EC (methylation) index (coefficients in Supplementary Data 2), which can distinguish normal from EC tissue (Fig. 4A). The perfect discrimination of normal tissue from EC tissue (AUC = 1) in both the training and hold-out testing sets indicated profound differences in DNA methylation patterns are present in EC that could be accurately detected using the TCGA-EC methylation index (Supplementary Fig. 4a).

We applied TCGA-EC methylation index to the endometrial biopsy data to compare the treatment groups. In the mifepristone group, a significant reduction in the TCGA-EC methylation index was observed post-

treatment compared to baseline ($P = 0.039$), whereas no significant change was detected in the control group ($P = 0.84$). When comparing the Δ post-pre-values between the mifepristone and control (vitamin B)-treated groups, a decrease in the EC methylation index was observed in the mifepristone-treated group, although this difference was not statistically significant ($P = 0.18$) (Fig. 4B, C). As we observed a significant increase in fibroblast proportion after mifepristone treatment, we further examined the TCGA-EC methylation index after accounting for both age and fibroblast proportions, which are known to substantially influence methylation patterns[45,46] (Supplementary Fig. 5). After this adjustment, no significant changes in the TCGA-EC methylation index were detected, indicating that, based on DNA methylation analysis, no cancer-associated changes were observed in the endometrium after mifepristone treatment.

### Gene expression profiling highlights mifepristone's impact on endometrial tissue

To provide a more comprehensive view of the biological changes in endometrial tissue after mifepristone treatment and their potential correlation with carcinogenesis, we extended our investigation by using RNA sequencing to explore alterations in gene expression. In parallel with the TCGA-EC methylation index, we used TCGA-UCEC gene expression data to derive

## Gene expression data

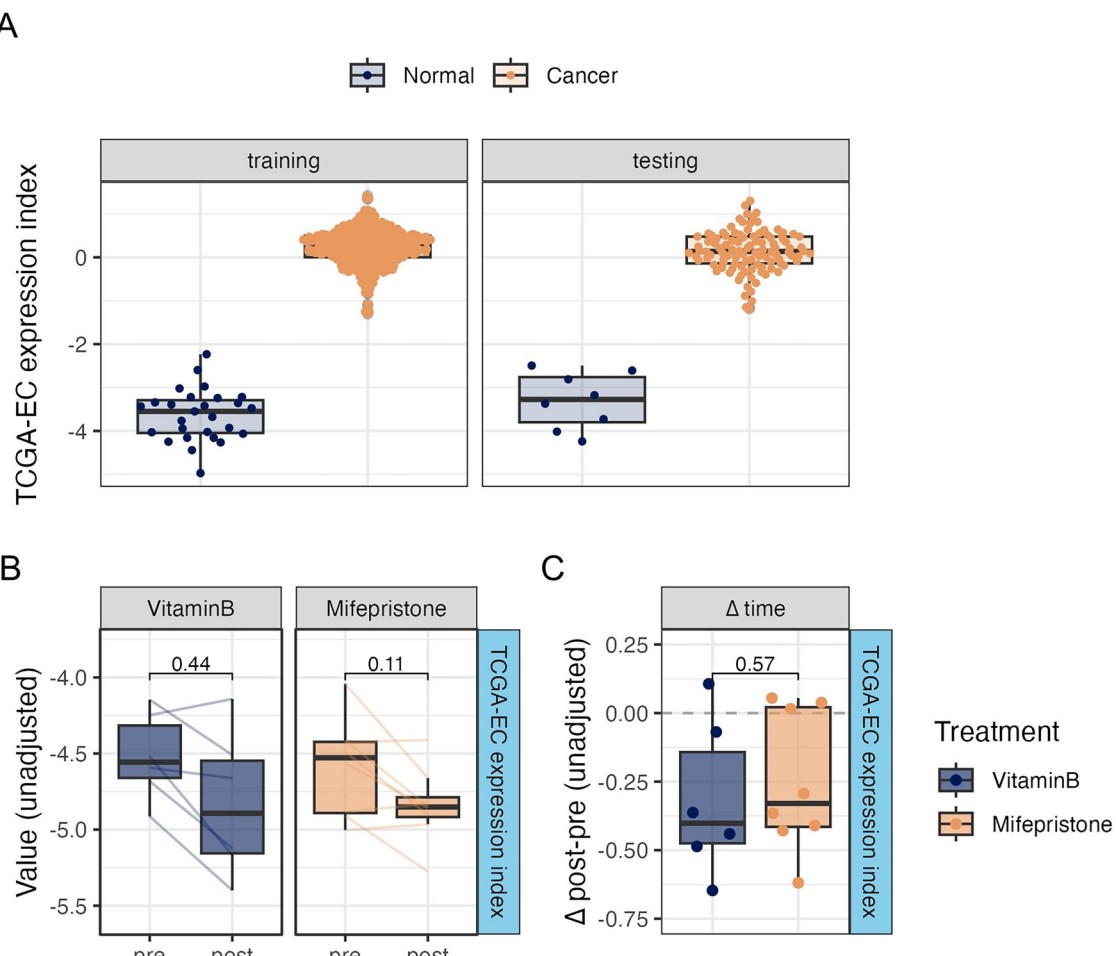

**Fig. 5 | Changes in cancer-indicative gene expression biomarkers upon mifepristone treatment. A** Expression-based index to distinguish normal from endometrial cancer tissue. Boxplots of the TCGA-EC expression index in the training, testing, and external validation sets (TCGA UCEC, GSE63678). **B** Paired pre and post values of cancer-indicative biomarkers associated with endometrial cancer based on RNA expression, trained using TCGA data (TCGA-EC score). *p* values are derived from paired Wilcoxon tests. **C** Change in biomarker values from baseline (Δ time) in Vitamin B and Mifepristone-treated individuals. *p* values derived from unpaired Wilcoxon tests comparing Mifepristone to Vitamin B. Sample sizes: *n* = 14 paired samples (mifepristone group: *n* = 8; control group: *n* = 6).

the TCGA-EC expression index. As observed for methylation, pronounced differences in gene expression were observed between the normal and control endometrial tissues. A total of 64 genes combined into the TCGA-EC expression index (coefficients in Supplementary Data 3) were able to distinguish well between normal and cancer tissue, with an AUC of 1 in the testing sets, and an AUC of 0.97 (95% CI: 0.89-1) in an external validation set (GSE63678, Supplementary Fig. 4b), indicating a high level of accuracy in distinguishing normal from cancerous tissues.

We tested the TCGA-EC expression index in our mifepristone-and control (vitamin B)-treated cohorts (Fig. 5B, C; Supplementary Fig. 5). Initially, no significant changes were observed when comparing the pre- and post-treatment samples within the mifepristone ($P = 0.11$) and control (vitamin B) groups ($P = 0.44$). Additionally, when comparing the Δ post-pre values between the vitamin B- and mifepristone-treated groups, no significant differences were observed ($P = 0.57$). However, after adjusting for age and fibroblast proportion, TCGA-EC expression index was significantly reduced after mifepristone treatment compared to that in the pre-treated samples ($P = 0.007$). The expression patterns of these genes were analyzed using TCGA-UCEC data, and we found no potential link to EC, as no distinct expression pattern was detected between EC tissue and the corresponding normal samples (Supplementary Fig. 6).

To further investigate whether mifepristone treatment influenced gene expression pathways related to endometrial carcinogenesis, we examined transcriptional shifts corresponding to three hallmarks of EC: early and late response to E, PI3K/AKT/mTOR, and KEGG-EC signaling. When comparing the post- and pre-treatment samples, no significant changes were observed in any of the hallmark gene sets. Additionally, adjusting for cell composition did not alter this outcome, further supporting that mifepristone treatment does not induce transcriptional changes linked to EC (Supplementary Fig. 7).

To examine the correlation between and impact of *BRCA* genes on EC, we developed and investigated a BRCA-EC gene signature. Using the R2 genomic analysis and visualization platform, we identified genes that correlated with *BRCA* and UCEC. No negative correlation was detected, and 89 positively correlated genes with an r value > 0.75 were selected for downstream analysis. We found that the expression levels of 79 out of 89 genes were significantly increased in EC compared to those in normal endometrial tissue (Supplementary Fig. 8a, b, and Supplementary Data 4). Testing the expression patterns of these genes and their correlation with mifepristone treatment showed that none of the genes were overexpressed after mifepristone treatment, three of which were significantly

downregulated, whereas the rest showed no significant alteration (supplementary Fig. 8c).

## Discussion

We systematically investigated the effects of mifepristone on endometrial tissue in women with *BRCA* PV mutations, focusing on its impact on endometrial safety, cellular composition, methylation patterns, and potential implications for endometrial carcinogenesis. These findings offer valuable insights into the mechanism of action of mifepristone and highlight the complex nature of hormonal modulation in the endometrial environment.

Consistent with previous findings, the dosage of mifepristone induced amenorrhea in all participants[38], likely due to the inhibition of ovulation[39]. No significant increase in endometrial thickness was detected post-treatment compared to that in the control (vitamin B)-treated group, suggesting no substantial proliferative response due to unopposed E effects. This aligns with earlier findings showing that mifepristone suppresses E-dependent mitotic activity in the endometrial glands; despite not binding to the ER, a phenomenon termed a functional noncompetitive anti-estrogenic effect[47].

To investigate this further, we analyzed the impact of mifepristone on the cellular composition of endometrial tissue and found that mifepristone treatment altered the cellular profile, notably reducing the proportion of epithelial cells while increasing fibroblast levels. This shift in cellular composition suggests that the observed endometrial thickening may be due to an increase in stromal or fibroblast cells, rather than epithelial proliferation. This distinction is crucial because epithelial cell proliferation is more closely associated with endometrial carcinogenesis, whereas an increase in fibroblast cells may indicate tissue remodeling and extracellular matrix expansion[48,49]. Importantly, this change did not translate into significant alterations in the immune cell landscape, suggesting a selective effect on fibroblast populations.

Additionally, the analysis of DNA methylation patterns provided insights into the effects of mifepristone on potential carcinogenesis in endometrial tissues. Using TCGA-EC methylation index, a validated biomarker that distinguishes between normal and cancerous endometrial tissues, we observed no significant alterations in the index following mifepristone treatment. This stability in TCGA-EC index suggests that mifepristone does not markedly increase the methylation-linked risk of endometrial carcinoma. Although a slight reduction in the index was noted, it was not statistically significant and likely represented a benign change in the methylation profiles. The role of DNA methylation in cancer development underscores the importance of this type of analysis[50]. DNA methylation of specific promoter regions across various cancer types suppresses gene expression, leading to irreversible gene silencing. This effectively locks stem or progenitor cells in an undifferentiated state, impairing cellular differentiation[51]. Thus, TCGA-EC methylation index is a valuable tool for assessing carcinogenic potential by capturing shifts in DNAme patterns, which are thought to arise from both genetic predispositions and exogenous factors[52]. Our findings reinforce the hypothesis that short-term mifepristone use exerts a minimal impact on methylation-driven carcinogenic pathways; however, long-term studies are required to confirm these observations.

Gene expression profiling offers further insights into the molecular changes induced by mifepristone. Using an expression-based TCGA-EC index, we found that this biomarker could accurately differentiate cancerous and normal tissues. When tested in mifepristone-treated endometrial samples, the expression index showed no significant increase post-treatment and there was no increase in the expression of genes linked to endometrial carcinogenesis. To further explore whether mifepristone induced oncogenic transcriptional changes, we examined specific gene expression hallmarks associated with EC risk. We found no significant alterations in the pathways linked to early and late E responses or PI3K/AKT/mTOR signaling, even after adjusting for cell composition. Similarly, no changes were observed in KEGG-EC signaling. provides a detailed examination of all the genes associated with *BRCA* signaling in EC. Our

analysis revealed no significant changes in gene expression profiles related to *BRCA* signaling after mifepristone treatment. Taken together, our study indicates no evidence of oncogenic molecular signatures after short-term mifepristone exposure. These findings should be interpreted as exploratory and limited to short-term treatment.

## Conclusion

Our findings provide further safety data that are important for future clinical development and implementation of PRMs. The results suggest no detectable molecular changes linked to endometrial carcinogenesis after short-term mifepristone exposure. However, long-term studies are essential to fully assess endometrial safety.

## Data availability

The raw data and processed data files were submitted to Gene Expression Omnibus database with the GEO accession ID GSE283656. Publicly available DNA methylation and gene expression data from TCGA-UCEC were accessed via TCGAbiolinks. The source data for the numerical figures is in Supplementary Data 5.

## Code availability

The codes for the analyses and figures are provided on GitHub (https://github.com/chiaraherzog/MifepristoneEndoMethyl), and at https://doi.org/10.5281/zenodo.18219939 [42].

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

## Acknowledgements

We would like to thank Eva Broberg, Anette Daberius, and Ulrika Fundin for their support in patient recruitment. This study was funded by the Swedish Research Council (2012-01981, 2017-00932), the Swedish Cancer Foundation (5321-9416), and a joint grant from the Stockholm County Council and Karolinska Institute (ALF funding).

## Author contributions

T.A.: Study design, data collection and acquisition, analysis, interpretation, writing, review, and editing. C.H.: Data curation and analysis, software, interpretation, review, and editing. M.F.R.: Data collection, writing, and manuscript review. N.R.B., D.U., and E.R. Data acquisition and curation. A.R. conceptualization, patient recruitment, and biopsy collection. M.W. and K.G.D. conceptualized, supervised, resources, wrote, reviewed, edited, and acquired the funding.

## Funding

## Competing interests

The authors declare no competing interests.

## Additional information

[1]Department of Women's and Children's Health, Karolinska Institutet Stockholm, Stockholm, Sweden. [2]European Translational Oncology Prevention and Screening Institute, Universität Innsbruck, Innsbruck, Austria. [3]Institute for Biomedical Ageing Research, Universität Innsbruck, Innsbruck, Austria. [4]UCL EGA Institute for Women's Health, University College London, London, UK. [5]Tirol Kliniken, Hall in Tirol, Austria. [6]Department of Pharmaceutical Basic Science, Faculty of Pharmacy, Tishk International University, Erbil, Iraq. [7]Division for Gynecology and Reproductive medicine, Karolinska University Hospital, Stockholm, Sweden. [8]Division of Hereditary Cancer, Theme Cancer, Karolinska University Hospital, Stockholm, Sweden. [9]General Directorate of Scientific Research Centre, Salahaddin University-Erbil, Erbil, Iraq. [10]These authors contributed equally: Martin Widschwendter, Chiara Herzog, Kristina Gemzell- Danielsson, Twana Alkasalias. ✉e-mail: twana.alkasalias@ki.se

