## [Transparent Peer Review file · Communications Medicine]

Short-term hormonal modulation with mifepristone does not induce oncogenic changes in the endometrium of BRCA1/2 mutation carriers

Corresponding Author: Dr Twana Alkasalias

Version 0:

Reviewer comments:

Reviewer #1

(Remarks to the Author)

This manuscript investigates the short-term endometrial safety of mifepristone, a progesterone receptor antagonist, in premenopausal women germline BRCA1/2 mutations. This is showcasing the secondary outcomes of a randomized, double-blind, placebo-controlled trial focused primarily in evaluating the potential breast cancer risk reduction using mifepristone. The trial was conducted over a span of three months, and participants received either mifepristone or a vitamin B placebo pill, administer every other day. Samples collected included matched endometrial biopsies, before and after the 90 days of the trial. Multi-omics analysis including transcriptomics (RNA-seq), DNA methylation profiling and cell-type deconvolution was employed to infer potential oncogenic changes caused by this drug.

The authors report that mifepristone caused amenorrhea in all treated participants without increasing endometrial thickness and showed no detectable epithelial proliferation or oncogenic molecular signatures. The authors conclude that short-term mifepristone exposure appears molecularly safe in this population.

The manuscript is scientifically valuable, but reframing the safety claims and addressing the methodological limitations is required before publication.

Below are the recommended changes needed:

Major:

1. Baseline biopsies for both vitamin B and mifepristone groups were collected at various menstrual phases. However, post-treatment biopsies, came from amenorrhic women. Since endometrial transcriptomic and methylation profiles can fluctuate significantly with hormonal/ovarian cycle changes, results should be carefully interpreted. While having matched samples (pre and post treatment) from the same patients strengthens this analysis, the lack of a standardized phase of the cycle for baseline biopsies can weaken the analysis as it will challenging to distinguish between changes linked to menstrual cycle stage and effects of the drug after treatment. Although the authors mention menstrual phase adjudgment in PCA, they should describe how phase variability was controlled statistically.

2. The methylation index was derived from TCGA UCEC data to infer methylation changes linked to endometrial carcinogenesis. However, this methylation index would be more reflective of advance endometrial tumors rather than early stage "oncogenic signatures". If this trial was designed to access long-term effects, this analysis would have been more beneficial to access such changes. It is recommended to rephrase conclusions to reflect the exploratory, preliminary nature of findings and limit claims to no evidence of oncogenic signatures rather than safety.

3. Three months is too brief to infer safety. Considering these are pre-menopausal women, it shouldn't be expected for oncogenic changes to appear within this short amount of time. Conclusions can be reframed to note this.

4. The population studied was women with germline BRCA1 and BRCA2 mutations carriers however, no additional information was provided about the distribution of mutations for each BRCA1 and BRCA2 and they were evenly distributed across control and experimental groups. Additional analyses were not done separately for each mutation which could mask additional biological insights.

Minor:

1. Since the trial assessed the "short-term" effect of mifepristone in the endometrial tissue, that should be reflected in the title.

2. Clarify the reason for the 50 mg every other day dosage of mifepristone. Although it is mentioned in the project plan attached briefly, it should be included in the body of the manuscript with any relevant citations as well.
3. A comprehensive analysis of hormone levels before, and after treatment with mifepristone would enhance the analysis.
4. The manuscript mentions PAEC however, no histopathological data was reported.
5. Ref 6 in introduction should be updated from 2018 to 2024/2025 trends.

Overall, the biological question is clinically relevant and would provide valuable insights into the safety of this drug in this high-risk population, however the interpretation of the results overreaches the data presented. The lack of evidence for oncogenic changes doesn't imply it is safe to use short-term.

Reviewer #2

(Remarks to the Author)

I co-reviewed this manuscript with one of the reviewers who provided the listed reports. This is part of the Communications Medicine initiative to facilitate training in peer review and to provide appropriate recognition for Early Career Researchers who co-review manuscripts.

Reviewer #3

(Remarks to the Author)

Reviewer Summary

This manuscript explores the relationship between BRCA1/2 pathogenic variants (PVs), hormonal regulation, and drug response. The topic is clinically important and relevant for understanding progesterone signaling and cancer risk, particularly in BRCA1 carriers. Importantly, the authors identify a key research gap—the studied drug has not been well characterized in the endometrium—making this investigation both novel and valuable. However, several aspects require clarification. It is unclear whether the samples represent healthy BRCA1/2 PV carriers or individuals previously diagnosed and currently disease-free. Although progesterone is emphasized, progesterone-targeted genes were not fully analyzed, limiting mechanistic insight. The authors also note that age and cell type affect DNA methylation but provide no citations (e.g., Torre, Vecchio & Greco, *Cells*, 2023; Smith, Hetzel & Meissner, *Nat Rev Genet*, 2025). Additional improvements include consistent p-value reporting and clarification that no significant changes were observed after accounting for age and fibroblast proportions.

Reviewer Comments

Overall Assessment

Overall, I found the manuscript to be clinically relevant and important for understanding the effects of this drug in the context of BRCA1/2 pathogenic variants (PVs). The study addresses an important gap in linking genetic predisposition with hormonal regulation, particularly the role of BRCA1 in progesterone signaling and its implications for triple-negative breast cancer (TNBC). Importantly, the manuscript also highlights that this drug has not been extensively studied in the endometrium, underscoring the novelty and clinical relevance of exploring its effects in this tissue context. However, several aspects of the study design, reporting, and interpretation require clarification and consistency to strengthen the manuscript's overall quality.

Major Comments

1. Clarification of Study Cohort

It is unclear whether the participants carrying BRCA1/2 pathogenic variants are (a) breast cancer-free carriers or (b) individuals with a prior breast cancer diagnosis who are currently free of disease. This distinction is crucial for interpreting the hormonal and molecular results, as the biological implications differ between unaffected carriers and survivors.

Recommendation: Please clarify explicitly whether your patient samples represent healthy BRCA1/2 PV carriers or individuals previously diagnosed with breast cancer who are disease-free at the time of sampling.

2. Progesterone-Targeted Gene Analysis

While the manuscript highlights progesterone levels as an important factor, no analyses were conducted on progesterone-responsive or progesterone-targeted genes. The pathway analysis instead focuses on endometrial cancer hallmark signatures (estrogen, PI3K/AKT/mTOR, and KEGG-EC pathways).

Recommendation: To support the stated significance of progesterone, consider including or discussing analyses of progesterone-regulated gene sets (e.g., PR signaling pathways or known PR target genes), or clarify why these were not assessed.

3. Lack of Citation Regarding Age and Cell Type Effects

The authors state that age and cell type are known to affect DNA methylation but do not provide supporting citations (lines 371-372).

Recommendation: Please include relevant references to support this statement. For example, see Torre, Vecchio & Greco (*Cells*, 2023) (<https://doi.org/10.3390/cells12081163>) and Smith, Hetzel & Meissner (*Nat Rev Genet*, 2025) (<https://doi.org/10.1038/s41576-024-00760-8>).

4. Consistency of Reported P-values

Some inconsistencies exist between the p-values reported in the text and those displayed in the figures. For instance, p-values noted as 0.007 in the Results section (for Figure 3 and Supplementary Figure 6) are listed as 0.0078 in the figures. These should be rounded consistently to 0.008.

Recommendation: Review all p-values across text, figures, and supplementary materials for consistent rounding and reporting.

5. Clarity of Sentence Structure and Comparative Statements

In one Results section, the sentence structure makes the comparison unclear. The text currently reads:

“...after accounting for both age group...”

This should be revised to:

“...after accounting for both age and fibroblast proportions.”

Additionally, the section should clearly conclude that no significant changes were observed after accounting for age and fibroblast proportions.

Recommendation: Revise this section for clarity and ensure that the comparative statements are explicit and grammatically correct.

Minor Comments

- A few sentences in the Results section require light proofreading and improved sentence structure for clarity and readability.

Version 1:

Reviewer comments:

Reviewer #1

(Remarks to the Author)

The key comments raised in the first round have been addressed, and the manuscript reads much more clearly now, particularly the added detail around cycle timing, dosing rationale, and the careful interpretation of results. These additions make the study easier to follow and more transparent.

It is also clear in the revision that the authors recognize this is a short-term evaluation and that the findings should be viewed as early molecular data rather than any statement about long-term safety. Given how new this area of investigation is, this small short-term trial is a reasonable starting point, and the manuscript now frames it that way.

With these revisions in place, I think the manuscript is in good shape and contributes useful early evidence to this field.

Reviewer #2

(Remarks to the Author)

I co-reviewed this manuscript with one of the reviewers who provided the listed reports. This is part of the Communications Medicine initiative to facilitate training in peer review and to provide appropriate recognition for Early Career Researchers who co-review manuscripts.

Reviewer #3

(Remarks to the Author)

The authors have responded to the critique appropriately.

Reviewer 1:

This manuscript investigates the short-term endometrial safety of mifepristone, a progesterone receptor antagonist, in premenopausal women germline BRCA1/2 mutations. This is showcasing the secondary outcomes of a randomized, double-blind, placebo-controlled trial focused primarily in evaluating the potential breast risk reduction using mifepristone. The trial was conducted over a span of three months, and participants received either mifepristone or a vitamin B placebo pill, administer every other day. Samples collected included matched endometrial biopsies, before and after the 90 days of the trial. Multi-omics analysis including transcriptomics (RNA-seq), DNA methylation profiling and cell-type deconvolution was employed to infer potential oncogenic changes caused by this.

The authors report that mifepristone caused amenorrhea in all treated participants without increasing endometrial thickness and showed no detectable epithelial proliferation or oncogenic molecular signatures. The authors conclude that short-term mifepristone exposure appears molecularly safe in this population.

The manuscript is scientifically valuable, but reframing the safety claims and addressing the methodological limitations is required before publication.

Below are the recommended changes needed:

Major:

1- Baseline biopsies for both vitamin B and mifepristone groups were collected at various menstrual phases. However, post-treatment biopsies came from amenorrheic women. Since endometrial transcriptomic and methylation profiles can fluctuate significantly with hormonal/ovarian cycle changes, results should be carefully interpreted. Although the authors mention menstrual phase adjudgment in PCA, they should describe how phase variability was controlled statistically.

We appreciate this important point. All baseline biopsies were collected during the luteal phase, as determined by cycle day at sampling. For post-treatment biopsies, all women treated with mifepristone were amenorrheic, whereas women in the vitamin B group continued menstruating; therefore, their post-treatment samples were also obtained during the luteal phase. This indicates that our baseline samples were standardized to the luteal phase rather than spanning multiple phases of the cycle.

To further assess whether methylation signatures could reflect menstrual cycle variation even within the luteal phase, we stratified baseline samples into early, mid, and late luteal sub-phases. No significant differences were observed across these sub-phases. As highlighted in Figure 2E, principal component analysis (PCA) confirmed that methylation profiles corresponded to menstrual phase, and all pre-treatment samples clustered within the luteal phase. Post-treatment samples from amenorrheic women represent a distinct hormonal state induced by mifepristone. We have clarified this in the Methods section.

2- The methylation index was derived from TCGA UCEC data to infer methylation changes linked to endometrial carcinogenesis. However, this methylation index would be more reflective of advanced endometrial tumors rather than early-stage “oncogenic signatures.” If this trial was designed to assess long-term effects, this analysis would have been more beneficial. It is recommended to rephrase conclusions to reflect the

exploratory, preliminary nature of findings and limit claims to no evidence of oncogenic signatures rather than safety.

We agree with the reviewer that the TCGA-derived methylation index cannot distinguish between “early” or “established” oncogenic processes. Our intention was to use this index as a sensitive biomarker to detect any molecular shift toward a cancer-like methylation profile, acknowledging that its predictive value for early transformation is limited.

In line with this recommendation, we have revised the Abstract, Discussion, and Conclusion to emphasize the exploratory nature of our findings.

3- Three months is too brief to infer safety. Considering these are pre-menopausal women, it shouldn't be expected for oncogenic changes to appear within this short amount of time. Conclusions can be reframed to note this.

We fully agree that a three-month duration is too short to draw definitive conclusions about endometrial safety, particularly in premenopausal women where oncogenic changes would not be expected to manifest within such a limited timeframe. Our intention was not to claim long-term safety but to provide preliminary molecular insights.

Importantly, we believe short-term evaluation is a necessary first step before initiating long-term studies. It would be ethically and scientifically challenging to start extended exposure without understanding the immediate molecular effects and potential risks. Furthermore, while clinical cancer outcomes require years to develop, the timing and nature of early molecular events that precede carcinogenesis remain poorly understood. Therefore, assessing whether any molecular alterations occur even after short-term treatment is of high value for risk stratification and trial design.

4- The population studied was women with germline BRCA1 and BRCA2 mutation carriers; however, no additional information was provided about the distribution of mutations for each BRCA1 and BRCA2, and whether they were evenly distributed across control and experimental groups. Additional analyses were not done separately for each mutation, which could mask additional biological insights.

Thank you for this observation. We have now included detailed information on the distribution of BRCA1 and BRCA2 mutations across the treatment groups in Supplementary Table 1. Specifically, among the participants with paired biopsies analyzed, the mifepristone group included 5 BRCA1 carriers and 3 BRCA2 carriers, while the vitamin B group included 2 BRCA1 carriers and 4 BRCA2 carriers.

The study was conducted as a randomized, double-blind, placebo-controlled trial, and allocation was performed blindly without consideration of mutation type. This design minimizes allocation bias and ensures comparability between groups.

We acknowledge that separate analyses for BRCA1 and BRCA2 carriers could provide additional biological insights. However, due to the sample size and the exploratory nature of this study, such stratified analyses were not feasible without compromising statistical power.

Patient	Treatment	Age	BRCA Status	Parity	Regular Menstruation (25-35 days)	BMI
1	Mifepristone	30	BRCA I	0	Yes	26.4
2	Mifepristone	30	BRCA II	0	Yes	26.3
3	Mifepristone	33	BRCA I	0	Yes	23.3
4	Mifepristone	35	BRCA II	2	Yes	22.6
5	Mifepristone	26	BRCA I	0	Yes	21.5
6	Mifepristone	39	BRCA II	3	Yes	21.9
7	Mifepristone	31	BRCA I	0	Yes	26.0
8	Mifepristone	42	BRCA I	1	Yes	31.6
9	Vitamin B	38	BRCA I	3	Yes	21.3
10	Vitamin B	41	BRCA I	1	Yes	20.1
11	Vitamin B	47	BRCA II	2	Yes	24.3
12	Vitamin B	31	BRCA II	0	Yes	21.4
13	Vitamin B	34	BRCA II	0	Yes	42.4
14	Vitamin B	25	BRCA II	0	Yes	19.9

Minor:

1. Since the trial assessed the “short-term” effect of mifepristone in the endometrial tissue, that should be reflected in the title.

Done (revised title): Short-term hormonal modulation with mifepristone does not induce oncogenic changes in the endometrium of BRCA1/2 mutation carriers.

2. Clarify the reason for the 50 mg every other day dosage of mifepristone. Although it is mentioned in the project plan attached briefly, it should be included in the body of the manuscript with any relevant citations as well.

Added to the method: The 50 mg every other day regimen was selected based on previous clinical studies demonstrating its efficacy in inducing amenorrhea without significant adverse effects (Refs 34, 38, 39). This dosing schedule also minimizes cumulative exposure while maintaining pharmacological activity.

3. A comprehensive analysis of hormone levels before, and after treatment with mifepristone would enhance the analysis.

Thank you for your insightful comment. While mifepristone is known to influence ovarian steroidogenesis, we did not measure endogenous hormone levels in this trial. This decision was intentional, as our primary objective was to investigate molecular changes in the endometrium rather than hormonal fluctuations. Furthermore, previous studies, including our earlier trial (Engman et al., 2008; DOI: 10.1093/humrep/dep100, Ref 38), have already confirmed the effects of mifepristone (50 mg every other day) on ovarian hormone levels.

4. The manuscript mentions PAEC however, no histopathological data was reported.

Thank you for pointing this out. Our analysis in the current study focused exclusively on molecular endpoints (DNA methylation and transcriptomics) rather than histopathological evaluation.

We also note that our previous studies have extensively examined histopathological changes associated with mifepristone and other PRMs, including PAEC, as highlighted in earlier publications (e.g., Engman et al., 2008; Mutter et al., 2008). Therefore, this trial was designed to complement those findings by investigating molecular changes in the endometrium.

5. Ref 6 in introduction should be updated from 2018 to 2024/2025 trends.

Thank you for this suggestion. We have updated the reference to reflect the most recent global epidemiology data on endometrial cancer (Bray et al.) and revised the corresponding sentence in the Introduction accordingly.

Overall, the biological question is clinically relevant and would provide valuable insights into the safety of this in this high-risk population, however the interpretation of the results overreaches the data presented. The lack of evidence for oncogenic changes doesn't imply it is safe to use short-term.

We appreciate this important observation and fully agree that the absence of oncogenic molecular signatures in our study does not equate to definitive short-term safety. Our intention was to report exploratory findings based on molecular endpoints rather than make clinical safety claims. To address this, we have revised the Abstract, Discussion, and Conclusion to ensure that the interpretation reflects the preliminary nature of the data and avoids overstatement.

Reviewer 2:

I co-reviewed this manuscript with one of the reviewers who provided the listed reports. This is part of the Communications Medicine initiative to facilitate training in peer review and to provide appropriate recognition for Early Career Researchers who co-review manuscripts.

We thank the reviewer for their contribution to the peer review process and appreciate their involvement in the Communications Medicine initiative. We note that no specific comments or revisions were requested.

Reviewer 3:

Reviewer Summary

This manuscript explores the relationship between BRCA1/2 pathogenic variants (PVs), hormonal regulation, and responses. The topic is clinically important and relevant for understanding progesterone signaling and risk, particularly in BRCA1 carriers. Importantly, the authors identify a key research gap—the studied has not been well characterized in the endometrium—making this investigation both novel and valuable. However, several aspects require clarification. It is unclear whether the samples represent healthy BRCA1/2 PV carriers or individuals previously diagnosed and currently disease-free. Although progesterone is emphasized, progesterone-targeted genes were not fully analyzed, limiting mechanistic insight. The authors also note that age and cell type affect DNA methylation but provide no citations (g., Torre, Vecchio & Greco, *Cells*, 2023; Smith, Hetzel & Meissner, *Nat Rev Genet*, 2025). Additional improvements include consistent p-value reporting and clarification that no significant changes were observed after accounting for age and fibroblast proportions.

Reviewer Comments

Overall Assessment

Overall, I found the manuscript to be clinically relevant and important for understanding the effects of this in the context of BRCA1/2 pathogenic variants (PVs). The study addresses an important gap in linking genetic predisposition with hormonal regulation, particularly the role of BRCA1 in progesterone signaling and its implications for triple-negative breast (TNBC). Importantly, the manuscript also highlights that this has not been extensively studied in the endometrium, underscoring the novelty and clinical relevance of exploring its effects in this tissue context. However, several aspects of the study design, reporting, and interpretation require clarification and consistency to strengthen the manuscript's overall quality.

Major Comments

1. Clarification of Study Cohort

It is unclear whether the participants carrying BRCA1/2 pathogenic variants are (a) breast-free carriers or (b) individuals with a prior breast diagnosis who are currently free of disease. This distinction is crucial for interpreting the hormonal and molecular results, as the biological implications differ between unaffected carriers and survivors.

Recommendation: Please clarify explicitly whether your patient samples represent healthy BRCA1/2 PV carriers or individuals previously diagnosed with breast who are disease-free at the time of sampling.

Thank you for highlighting this important point. All participants were healthy BRCA1/2 pathogenic variant carriers without a prior mastectomy nor diagnosis of breast cancer or other malignancies. This was explicitly ensured during screening, which included a

comprehensive medical history and clinical examination. We will clarify this in the Patients and Methods section by adding:
“Participants were unaffected BRCA1/2 PV carriers with no previous history of risk-reducing mastectomy or diagnosis of breast or other cancers.”

2. Progesterone-Targeted Gene Analysis

While the manuscript highlights progesterone levels as an important factor, no analyses were conducted on progesterone-responsive or progesterone-targeted genes. The pathway analysis instead focuses on endometrial hallmark signatures (estrogen, PI3K/AKT/mTOR, and KEGG-EC pathways).

Recommendation: To support the stated significance of progesterone, consider including or discussing analyses of progesterone-regulated gene sets (g., PR signaling pathways or known PR target genes), or clarify why these were not assessed.

Thank you for this important suggestion. Our primary aim was to assess endometrial safety and cancer-related molecular signatures rather than progesterone signaling per se. However, we agree that PR-target genes are relevant for mechanistic insight. To address this, we have now included an exploratory analysis of a previously described Progesterone Receptor Binding Site (PRBS) signature in our dataset (DOI: [10.1038/s41467-021-27918-w](https://doi.org/10.1038/s41467-021-27918-w), DOI: [10.1038/s43856-025-00779-w](https://doi.org/10.1038/s43856-025-00779-w)). This signature reflects genes with known PR binding sites and is commonly used to approximate progesterone responsiveness.

We evaluated PRBS scores before and after treatment in both groups (Vitamin B and mifepristone), using paired comparisons and adjustments for cell-type composition. As shown in the following plots, no significant changes were observed in PRBS scores following mifepristone treatment, either in unadjusted or cell composition-adjusted analyses. These findings suggest that short-term mifepristone exposure does not significantly alter the expression of PR-target gene sets in endometrial tissue.

Progesterone plays a critical role in opposing estradiol-driven proliferation in the endometrium; when this opposition is lost, unopposed estrogen can exert additional proliferative effects, increasing the risk of endometrial hyperplasia and potentially endometrial cancer. Therefore, endometrial safety remains one of the key considerations in the clinical implementation of progesterone receptor modulators (PRMs). It is critical to assess the actual impact of PRMs on endometrial proliferation and carcinogenesis. Our findings contribute important safety data that will inform future development and long-term evaluation of PRMs in high-risk populations such as BRCA1/2 carriers.

Supported Figure 1: Changes in Progesterone Receptor Binding Site (PRBS) signature scores before and after treatment with Vitamin B or mifepristone. (a) Unadjusted values; (b) Values adjusted for cell-type composition. No significant differences were observed in either analysis.

3. Lack of Citation Regarding Age and Cell Type Effects

The authors state that age and cell type are known to affect DNA methylation but do not provide supporting citations (lines 371-372).

Recommendation: Please include relevant references to support this statement. For example, see Torre, Vecchio & Greco (Cells, 2023) (<https://doi.org/10.3390/cells12081163>) and Smith, Hetzel & Meissner (Nat Rev Genet, 2025) (<https://doi.org/10.1038/s41576-024-00760-8>).

Thank you for pointing this out. We agree that this statement requires supporting references. Age and cell type composition are well-established factors influencing DNA methylation patterns. We have now added the following citations to substantiate this claim:

- Torre, Vecchio & Greco, Cells (2023): <https://doi.org/10.3390/cells12081163>
- Smith, Hetzel & Meissner, Nat Rev Genet (2025): <https://doi.org/10.1038/s41576-024-00760-8>

4. Consistency of Reported P-values

Some inconsistencies exist between the p-values reported in the text and those displayed in the figures. For instance, p-values noted as 0.007 in the Results section (for Figure 3 and Supplementary Figure 6) are listed as 0.0078 in the figures. These should be rounded consistently to 0.008.

Recommendation: Review all p-values across text, figures, and supplementary materials for consistent rounding and reporting.

Thank you for noting this. We have reviewed all p-values across the text, figures, and supplementary materials. To maintain accuracy and consistency, we now report p-values exactly as calculated (e.g., 0.0078) in both the text and figures, rather than rounding. This approach ensures transparency and precision in statistical reporting.

5. Clarity of Sentence Structure and Comparative Statements

In one Results section, the sentence structure makes the comparison unclear. The text currently reads:

“...after accounting for both age group...”

This should be revised to:

“...after accounting for both age and fibroblast proportions.”

Additionally, the section should clearly conclude that no significant changes were observed after accounting for age and fibroblast proportions.

Recommendation: Revise this section for clarity and ensure that the comparative statements are explicit and grammatically correct.

Thank you for pointing this out. We have revised the sentence for clarity and explicitly stated the outcome. The revised text now reads:

“As we observed a significant increase in fibroblast proportion after mifepristone treatment, we further examined the TCGA-EC methylation index after accounting for both age and fibroblast proportions, which are known to substantially influence methylation patterns (Supplementary Figure 5). After this adjustment, no significant changes in the TCGA-EC methylation index were detected, suggesting that from a

DNA methylation perspective, no cancer-associated changes were observed in the endometrium after mifepristone treatment.”

Minor Comments

- A few sentences in the Results section require light proofreading and improved sentence structure for clarity and readability.

Thank you for this suggestion. We have carefully reviewed the Results section and made minor edits to improve clarity and readability without altering the scientific meaning. Examples of changes include:

Original: The primary finding was that all the participants (100%) in the mifepristone group experienced amenorrhea during the follow-up visit.

Revised: All participants (100%) in the mifepristone group experienced amenorrhea during follow-up.

Original: ...although this did not reach statistical significance ($P=0.078$), along with a significant increase in the proportion of fibroblasts (from 31% to 36%, $P=0.007$).

Revised: ...although this decrease was not statistically significant ($P=0.078$), fibroblast proportion increased significantly from 31% to 36% ($P=0.0078$).

Reviewer 1

The key comments raised in the first round have been addressed, and the manuscript reads much more clearly now, particularly the added detail around cycle timing, dosing rationale, and the careful interpretation of results. These additions make the study easier to follow and more transparent. It is also clear in the revision that the authors recognize this is a short-term evaluation and that the findings should be viewed as early molecular data rather than any statement about long-term safety. Given how new this area of investigation is, this small short-term trial is a reasonable starting point, and the manuscript now frames it that way. With these revisions in place, I think the manuscript is in good shape and contributes useful early evidence to this field.

Response:

We thank the reviewer for the positive assessment and for acknowledging the improvements made in clarity and interpretation. We have maintained this framing throughout the manuscript and ensured that the limitations regarding short-term evaluation and the need for long-term studies are clearly stated in the Abstract, Introduction, and Discussion.

Reviewer 2

I co-reviewed this manuscript with one of the reviewers who provided the listed reports. This is part of the Communications Medicine initiative to facilitate training in peer review and to provide appropriate recognition for Early Career Researchers who co-review manuscripts.

Response:

We appreciate the reviewer's contribution. Thank you for your time and effort in evaluating our work.

Reviewer 3

The authors have responded to the critique appropriately.

Response:

We thank the reviewer for confirming that our responses and revisions adequately addressed the previous concerns.